

# Novel grey wolf optimizer based parameters selection for GARCH and ARIMA models for stock price prediction

Sneha S. Bagalkot[1,2], Dinesha H. A[1,3] and Nagaraj Naik[4]

[1] Nagarjuna College of Engineering and Technology, Bengaluru and Visvesvaraya Technological University, Belagavi, India
[2] B.M.S. College of Engineering, Bengaluru, India
[3] SIET, Tumkur, Karnataka, India
[4] Computer Science & Engineering, Manipal Institute of Technology, Manipal Academy of Higher Education (MAHE), Manipal, Karnataka, India

## ABSTRACT

Stock price data often exhibit nonlinear patterns and dynamics in nature. The parameter selection in generalized autoregressive conditional heteroskedasticity (GARCH) and autoregressive integrated moving average (ARIMA) models is challenging due to stock price volatility. Most studies examined the manual method for parameter selection in GARCH and ARIMA models. These procedures are time-consuming and based on trial and error. To overcome this, we considered a GWO method for finding the optimal parameters in GARCH and ARIMA models. The motivation behind considering the grey wolf optimizer (GWO) is one of the popular methods for parameter optimization. The novel GWO-based parameters selection approach for GARCH and ARIMA models aims to improve stock price prediction accuracy by optimizing the parameters of ARIMA and GARCH models. The hierarchical structure of GWO comprises four distinct categories: alpha ($\alpha$), beta ($\beta$), delta ($\delta$) and omega ($\omega$). The predatory conduct of wolves primarily encompasses the act of pursuing and closing in on the prey, tracing the movements of the prey, and ultimately launching an attack on the prey. In the proposed context, attacking prey is a selection of the best parameters for GARCH and ARIMA models. The GWO algorithm iteratively updates the positions of wolves to provide potential solutions in the search space in GARCH and ARIMA models. The proposed model is evaluated using root mean squared error (RMSE), mean squared error (MSE), and mean absolute error (MAE). The GWO-based parameter selection for GARCH and ARIMA improves the performance of the model by 5% to 8% compared to existing traditional GARCH and ARIMA models.

# INTRODUCTION

The prediction of stock prices has always been challenging due to the dynamic and complex nature of financial markets (*Kehinde, Chan & Chung, 2023*; *Sheth & Shah, 2023*).

Corresponding author
Nagaraj Naik, nagaraj.naik@manipal.edu

Researchers and analysts in the financial industry have been looking for ways to increase the reliability of stock price prediction models (*Chudziak, 2023*; *Han, Kim & Enke, 2023*). However, it is difficult to predict stock prices due to the global market fluctuation. Autoregressive integrated moving average (ARIMA) and generalized autoregressive conditional heteroskedasticity (GARCH) models have been developed to predict stock prices accurately (*Sisodia et al., 2022*; *Yadav, Yadav & Saini, 2022*). The ARIMA and GARCH model's primary concern is that selecting parameters is essential in reducing forecasting errors. However, parameter selection in ARIMA and GARCH models is a challenging task because most of the work considered a manual method to select the parameters in ARIMA and GARCH models (*Zhao et al., 2022*; *Dhafer et al., 2022*). In order to improve the accuracy of stock price predictions, we considered the grey wolf optimizer (GWO) method to select the best parameters for GARCH and ARIMA models.

The volatility typically seen in stock prices, the GARCH model is widely used for modeling volatility and forecasting stock prices (*Mahajan, Thakan & Malik, 2022*; *Zeghdoudi, Lallouche & Remita, 2014*). The prediction performance of the GARCH model is very sensitive to the accuracy with which its parameters are chosen (*Fatima & Uddin, 2022*; *Brooks & Burke, 2003*). The volatility or uncertainty in asset values is a crucial aspect of stock price analysis (*Sen, Mehtab & Dutta, 2021*). Because of the complexity and ability to capture volatility, clustering is complex. How much previous volatility is carried over into projections of future volatility is a function of the parameters chosen for GARCH models (*Chou, 1988*; *Hong et al., 2023*). Investors can make more accurate risk management, option pricing, and portfolio optimization decisions with the help of GARCH models when they use well-chosen parameters to estimate and forecast volatility (*Chu & Freund, 1996*; *Molnár, 2016*). The ARIMA model is widely used for time series forecasting since it provides for both autoregressive and moving average properties (*Shah, Bhatt & Shah, 2022*; *Merabet & Zeghdoud, 2020*). Effective time series modeling relies heavily on the careful selection of ARIMA model parameters, such as the order of autoregressive (p), integrated (d), and moving average (q) terms (*Poddar et al., 2020*; *Kumar, Kumar & Kumar, 2022*). Poor data modeling due to inaccurate parameter selection can lead to inaccurate forecasts and interpretations. ARIMA models can improve prediction accuracy by capturing underlying patterns, seasonality, and cyclicity, provided the correct parameters are used. Parameter selection in ARIMA and GARCH models is critical due to higher stock price volatility. Therefore, the GWO method is considered in this work to select parameters in ARIMA and GARCH models.

The GWO is a metaheuristic approach that takes its cues from the social structure of wild grey wolves (*Mirjalili, Mirjalili & Lewis, 2014*; *Faris et al., 2018*). It mimics a wolf pack's hunting and leadership structure to identify the best solution (*Mirjalili et al., 2016*; *Gupta & Deep, 2019*). To accomplish this task, it mimics the wolves' behaviors, such as searching the prey, encircling, and attacking the prey. The GWO offers a robust optimization framework for dealing with complicated issues by emulating these behaviors (*Makhadmeh et al., 2023*). An innovative strategy for parameter selection for GARCH and ARIMA models using the GWO algorithm has not been studied in the literature. This work aims to minimize the forecast error by using the GWO method. The GWO-GARCH and GWO-ARIMA model

aims to improve stock price predictions' accuracy by eliminating the need for human intervention in the parameter selection process.

The primary work of this paper is as follows:

- Grey wolf optimizer is considered to select the best parameters in ARIMA and GARCH models.
- GWO-ARIMA and GWO-GARCH method is used to predict the stock prices.

The Literature reviews section discusses the literature reviews, and the methodology is discussed in the Methodology section. Results analysis presented in the Results analysis section and the Conclusions section concludes the proposed work.

## LITERATURE REVIEWS

*Fang, Lee & Su (2020)* presented the generalized information criteria (GIC) method to ascertain the tuning parameter. It helps select the best model for a given dataset by balancing the goodness of fit and the model's complexity. The GIC comprises two distinct components. The first component measures data compliance, while the second measures model intricacy. The study suggests that GIC operates on a trade-off basis between the accuracy of the model's fit and its level of complexity.

*Joyo & Lefen (2019)* study explores Pakistan's equities markets' interrelationships and portfolio diversification with its partners, including China and the US. The study examined Pakistan's and its trading partners' stock market correlation and volatility using the dynamic conditional covariance (DCC) technique.

*Sun & Yu (2020)* discussed support vector regression (SVR) and GARCH models to forecast volatility in the stock. As an alternative to the SVR-GARCH method, this GARCH-SVR method is proposed, which uses the SVR estimation technique to estimate the GARCH parameters in place of the maximum likelihood estimation. Asymmetric volatility effects are not able to be captured in GARCH-SVR models. The study considered S&P 500 index data.

*Zolfaghari & Gholami (2021)* considered GARCH and ARIMA models for stock price forecasting. First, estimate the ARMA (p, q) model and residual diagnostic tests. Second, using the autocorrelation (AC) and partial autocorrelation (PAC) functions, vary "p" and "q" values from 0 to 6 and find the best-fitted model using the BIC. Third, GARCH family model-based conditional variance estimation.

*Kumar Chandar (2021)* presented Elman neural network (ENN)-based stock price prediction. ENN parameter settings are usually determined *via* trial and error. This study optimizes ENN parameters with grey wolf optimization GWO. Later, optimized parameters were given into ENN for stock price prediction.

*Sivaram et al. (2020)* considered optimal least square support vector machine (OLS-SVM) to estimate Blockchain financial product return rates. GWO and differential evolution (DE) methods were used to finetune the parameters. Combining the GWO and DE methods helps reduce GWO's local optima and increases population diversity. The experimental results were analyzed using MSE and MAPE.

*Chun et al. (2021)* developed a stock forecasting model by considering investor emotions. Microblogging counts the frequency of emotional terms and documents the frequency captured. Adjectives, nouns, adverbs, and interjections are extracted using POS tagging from micro-blogging data. The retrieved POS is classified using emotions such as joy and sadness.

Few studies have demonstrated that Elman neural network (ENN) is well-suited for financial market forecasting because of its feedback link, local structure, and greater capacity to handle dynamic input (*Kumar Chandar, 2021*). ENN is built on the backpropagation feed forward neural network (BPNN). Due to using the BP algorithm for weight optimization, ENN suffers from certain drawbacks, such as local minima and delayed convergence. The related work is shown in Table 1.

Several statistical domains have recently seen soft computing techniques like ANN and fuzzy logic for financial market prediction (*Chun et al., 2021*). Financial market forecasting is an important area of study. *Atsalakis & Valavanis (2009)* found that different ANN models predict the stock market. Because of its inherent non-linearity, self-study, self-adaptation, related memory, and self-organization, artificial neural networks (ANN) have successfully predicted stock market data. ANN can learn from input samples and extract hidden information if the functional relationships are challenging to identify patterns.

Most of the studies considered GARCH and ARIMA models for stock price prediction. The manual method has been considered for parameter selection in GARCH and ARIMA models in literature work. These methods are trial and error as well as time-consuming. To overcome this, we have considered a GWO approach for selecting the best parameters in GARCH and ARIMA models. GWO's population-based search method simultaneously searches various search spaces to improve the effectiveness of solutions.

## METHODOLOGY

Stock data are collected from the National Stock Exchange (NSE), India. Axis Bank, HFDC Bank, Infosys, TCS, SBIN, and Adani stock are considered for experimental work. The dataset encompasses a significant period, including 2008 and 2023. The stock data contains four variables: open price, high price, low price, and close price.

In this work, we have identified the volatility in the stock price. Volatility helps traders and investors evaluate stock risk (*Laurent & Shi, 2020*). Price volatility increases returns and losses to traders. In addition, volatility may affect trading approaches and investing choices. Investors focus on volatility trading to capitalize on market fluctuations by buying low and selling high. Therefore, understanding volatility is essential for risk management and investing. Historical price data is used to calculate volatility using standard deviation. The standard deviation of the logarithmic returns of stock prices over a given period is computed. A more minor standard deviation indicates low volatility, while a larger one implies high volatility. The stock price and its volatility are described in Figs. 1 and 2.

To identify cluster volatility, the proposed work considered GARCH and ARIMA models. The ARIMA model can analyze and predict time series data. It analyses data patterns using autoregressive (AR), differencing (I), and moving average (MA) components. The

**Table 1 Literature reviews on stock price prediction.**

| Author | Data | Method | Outcome | Merit | Remarks |
|---|---|---|---|---|---|
| *Fang, Lee & Su (2020)* | S&P500 | GARCH-MIDA | Stock price volatility prediction | The variable selection GARCH-MIDAS model predicts long-term stock market volatility. | Work considered GARCH(1,1) parameter estimation. |
| *Joyo & Lefen (2019)* | Pakistan Stock Exchange | DCC-GARCH | Volatility estimation | This approach measures volatility and correlation at every time, which helps identify shock news. | It increases the computational complexity when dealing with a sizeable high-frequency dataset. |
| *Sun & Yu (2020)* | S&P 500 and GBP/USD exchange | GARCH-SVR | S&P 500 stock returns prediction | The SVR-GARC model performs well with and without financial crises. | Due the integration of the two models, it creates complexity in model selection and parameter tuning. |
| *Zolfaghari & Gholami (2021)* | Dow Jones Data | ARIMA-GARCH | Stock index prediction | In-sample findings showed that the ARIMA-GARCH model fitted well for stock index prediction. | It can be computationally expensive and require different optimization approaches to estimate the parameters of both components. |
| *Kumar Chandar (2021)* | NASDAQ stock data | GWO-Elman neural network | Stock price prediction | Experimental and statistical results show that the GWO method outperformed the traditional method. | The performance of the GWO method is sensitive to the selection of parameters. |
| *Bazrkar & Hosseini (2023)* | S&P 500 | Particle Swarm Optimization(PSO) | Stock price prediction | Particle Swarm Optimisation (PSO) demonstrates a notable proficiency in identifying global optima within a given search field. | Overfitting is possible in PSO if the parameters are incorrectly set or the model complexity needs to be managed. |
| *Abdual-Salam, Abdul-Kader & Abdel-Wahed (2010)* | Poland | Differential Evolution (DE) and PSO | Stock price prediction | Differential equations (DE) can handle parameter spaces that are both continuous and discrete. | It optimizes the model's parameters but does not explain why or how they relate to stock market movements. |
| *Chung & Shin (2020)* | Korea Composite Stock Price Index | Genetic Algorithim Optimization | Stock price prediction | GA is capable of handling both continuous and discrete parameter spaces. | Parameter finetuning, such as population size, mutation rate, and crossover rate, is difficult in GA. |

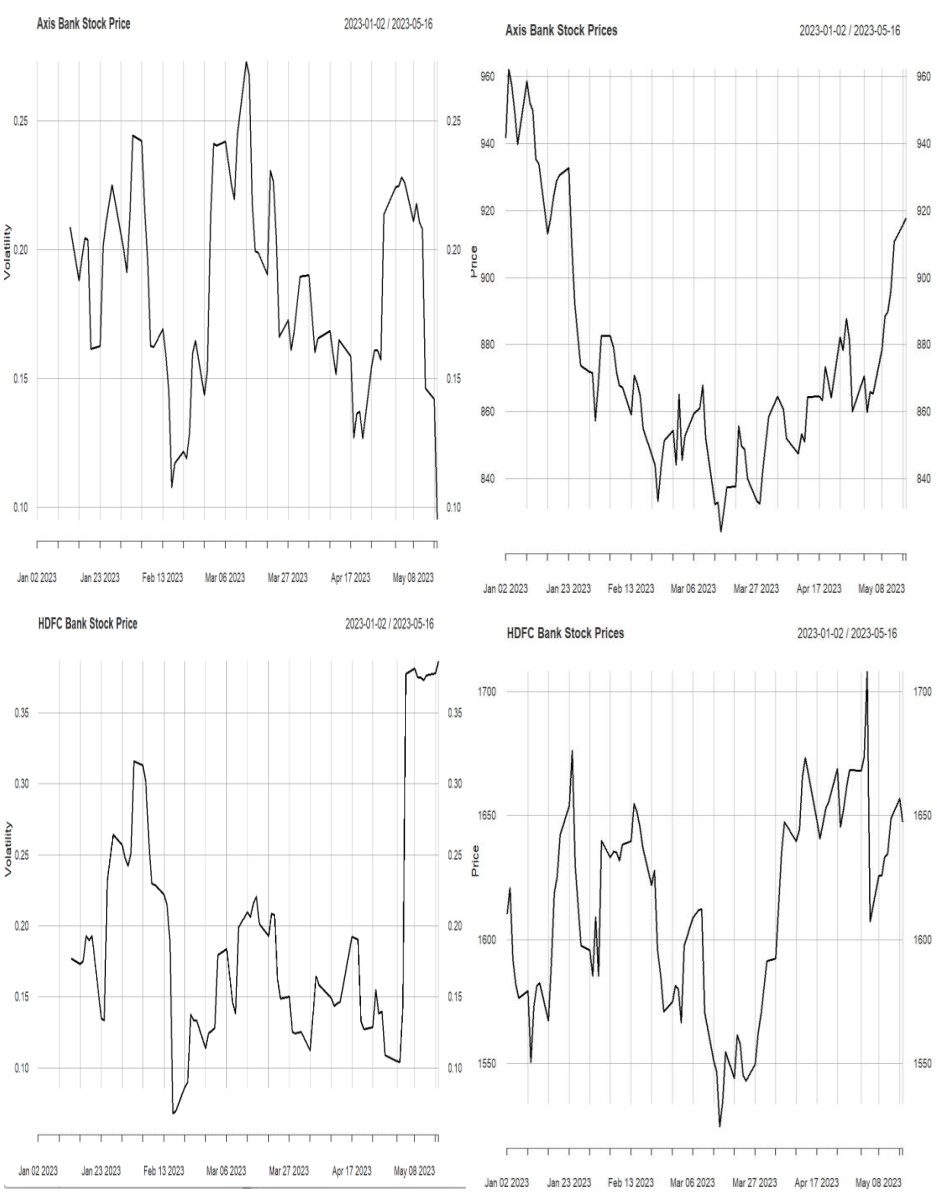

**Figure 1  Stock price and volatility.**

ARIMA model's AR component captures the linear relationship between current and prior observations. The lag order, "p," defines the number of lagged terms in the model. The AR component captures long-term patterns and series persistence. ARIMA differencing accounts for time series non-stationarity. Differentiating the series removes trends and seasonality, making the data stationary. Stationarity depends on "d," the differencing order. ARIMA's MA component models differenced series residual errors. It captures short-term data variations or random shocks. The MA component's lag order is "q", showing the model's lagged mistakes.

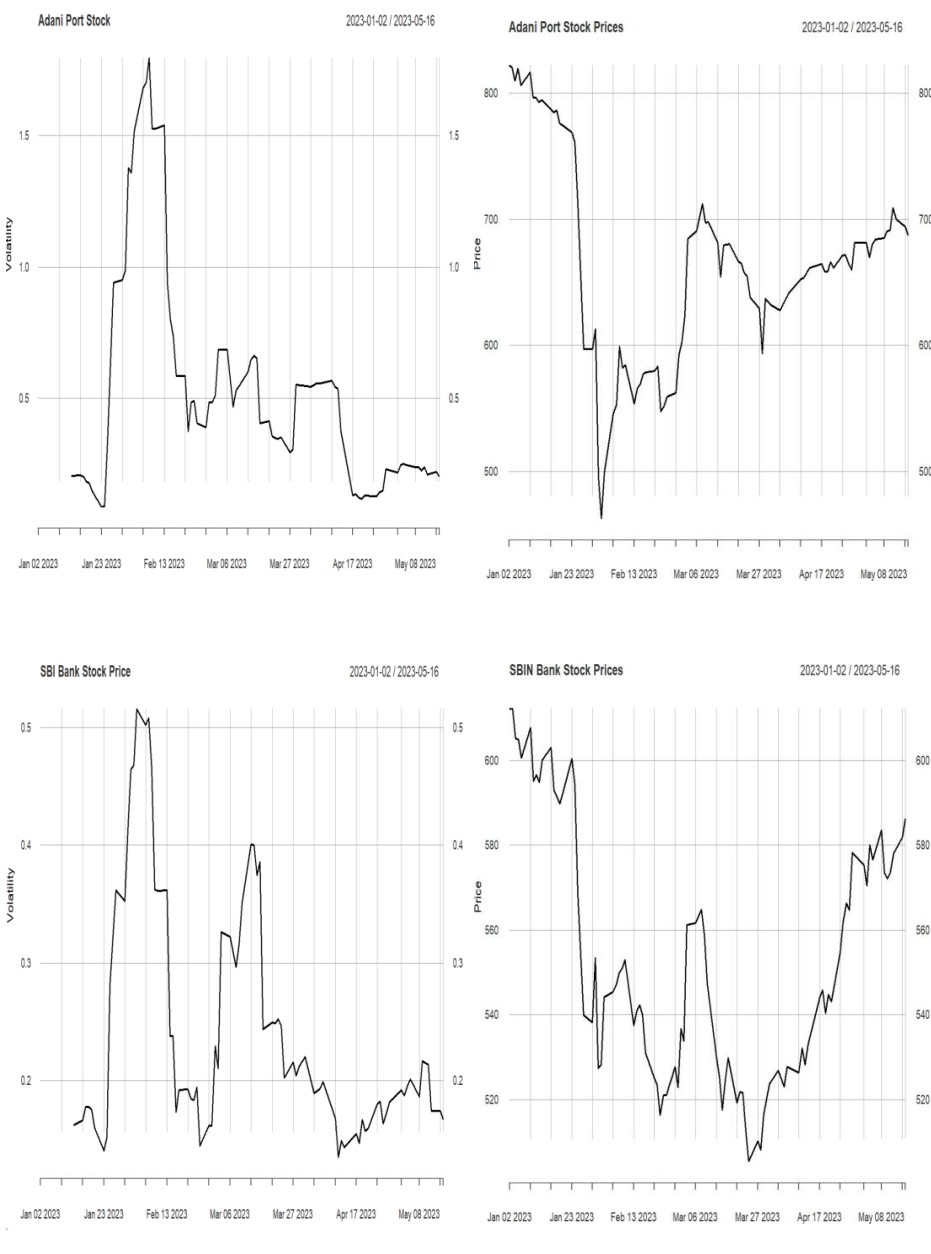

**Figure 2  Stock price and volatility.**

In the proposed work, we found a high variance in volatility clustering. Therefore, we considered the GARCH model. The GARCH model accounts for financial data's time-varying volatility. It captures conditional heteroskedasticity, where variance changes with past observations. The autoregressive (AR) and moving average (MA) terms on squared residuals make up the GARCH model. However, in GARCH and ARIMA models, parameter selection is carried out using trial and error, and it is a challenging and time-consuming

task. Therefore, the proposed work considered the GWO method for parameter selection in ARIMA and GARCH models.

The GWO is an optimization algorithm inspired by nature and designed to behave like grey wolves regarding their social structure and hunting behavior (*Rezaei, Bozorg-Haddad & Chu, 2018*; *Mirjalili, Mirjalili & Lewis, 2014*). It is a population-based metaheuristic method that has shown effectiveness in solving various optimization problems, including medical diagnosis and classification. The hierarchical structure of grey wolves' social organization comprises four distinct categories: alpha ($\alpha$), beta ($\beta$), delta ($\delta$), and omega ($\omega$). The predatory conduct of wolves primarily encompasses the act of pursuing and closing in on the prey, tracing the movements of the prey, and ultimately launching an attack on the prey. In the proposed, attacking prey is a selection of the best parameters for GARCH and ARIMA models. The overall workflow is described in Fig. 3.

## GWO method for parameter selection in ARIMA and GARCH model

In the GWO method, the population of candidate solutions is represented by a pack of "wolves", which communicate with one another and adjust their locations based on the actions of the pack's alpha, beta, and delta wolves. In this article, the fitness levels of the wolves are used to classify them as alpha, beta, or delta. The wolf with the highest fitness is chosen as the alpha wolf, the second-best wolf is the beta wolf, and the third-best wolf is the delta wolf. These wolves are essential because they help direct the search and shape the updates the other wolves receive. MSE and RMSE metric is considered to calculate the fitness value. The proposed work GWO-based parameter selection for ARIMA and GARCH model steps are described in Algorithm 1. The selection of parameters for ARIMA and GARCH using GWO is as follows. The first is to initialize the population of wolves with random positions using random numbers. Each wolf represents a candidate solution, and the aim is to identify the optimal solution. In the second step, define the initial value for $a = 2$ and encircle the prey. The third is to evaluate each wolf's fitness by applying the RMSE and MSE metrics to its corresponding position. The fitness represents the quality of the solution. The fourth is to identify the wolves with the first-highest, second-highest, and third-highest wolves using the fitness values of wolves. The fifth is to update the positions of the remaining wolves. The sixth is to select the best solution by taking the average of three wolves. The positions are adjusted to explore the solution space and converge toward better solutions.

In this expression, t represents the current iteration, $\vec{A}$ and $\vec{C}$ are coefficient vectors, $\vec{G_p}$ is the prey's position vector, $\vec{G}$ shows the grey wolf's position vector. The variable "D" represents the distance between the grey wolf and its prey, and it is defined in Eqs. (1) and (2). Iteratively reducing the number of components in $\vec{a}$ from 2 to 0 while generating random $\vec{r1}$ and $\vec{r2}$ in the interval [0, 1] and it is defined in Eqs. (3) and (4).

$$\vec{D} = |\vec{C}.\overrightarrow{G_{p(t)}} - \overrightarrow{G_{(t)}}| \tag{1}$$

$$\vec{G}(t+1) = \vec{G}_{p(t)} - \vec{A}.\vec{D} \tag{2}$$

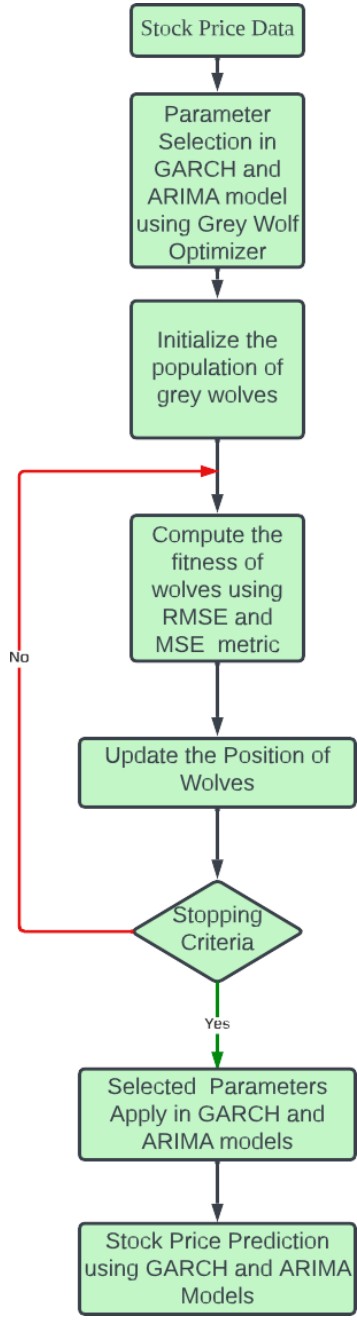

**Figure 3  Flow of proposed work.**

---

**Algorithm 1** GWO-based parameter selection for ARIMA and GARCH model algorithm

---

1:  The population of grey wolves Gi ($i = 1,2,\ldots,n$) is created in a random manner.

2:  Assign the initial value of the variable "a" as 2 using $\vec{A} = 2\vec{a}\,\vec{r1} - \vec{a1}$ and $\vec{C} = 2\vec{r}_2$

3:  Compute the fitness level of every individual within the population using RMSE, MSE, and MAE metrics for below wolves.
$$\vec{G1} = \vec{G\alpha} - \vec{A1}.\vec{D\alpha}$$
$$\vec{G2} = \vec{G\beta} - \vec{A2}.\vec{D\beta}$$
$$\vec{G3} = \vec{G\delta} - \vec{A3}.\vec{D\delta}$$

4:  Identify the wolves with the highest, second-highest, and third-highest fitness values, respectively, as the alpha, beta, and delta wolves.

5:  For the range of values of t from 1 to the maximum number of iterations, update the parameters:
$$\vec{D_\alpha} = \left| \vec{C1}.\vec{G_\alpha} - \vec{G} \right|$$
$$\vec{D_\beta} = \left| \vec{C2}.\vec{G_\beta} - \vec{G} \right|$$
$$\vec{D_\delta} = \left| \vec{C3}.\vec{G_\delta} - \vec{G} \right|$$

6:  Select the best solution by taking the average of 3 wolves. $\vec{G}(t+1) = \frac{\vec{G1} + \vec{G2} + \vec{G3}}{3}$

---

$$\vec{A} = 2\vec{a}\,\vec{r1} - \vec{a1} \tag{3}$$

$$\vec{C} = 2\vec{r}_2 \tag{4}$$

Where r1 and r2 are random variables between zero and one, and a convergence factor falls from two to zero as n iterations increase. The alpha, beta, and delta wolves direct the omega wolves in their pursuit of the prey, and the omega wolves recalculate the prey's location based on the best estimates of the alpha, beta, and delta wolves. The pack of grey wolves is positioned as shown by Eqs. (5) to (11).

$$\vec{D_\alpha} = \left| \vec{C1}.\vec{G_\alpha} - \vec{G} \right| \tag{5}$$

$$\vec{D_\beta} = \left| \vec{C2}.\vec{G_\beta} - \vec{G} \right| \tag{6}$$

$$\vec{D_\delta} = \left| \vec{C3}.\vec{G_\delta} - \vec{G} \right| \tag{7}$$

$$\vec{G1} = \vec{G\alpha} - \vec{A1}.\vec{D\alpha} \tag{8}$$

$$\vec{G2} = \vec{G\beta} - \vec{A2}.\vec{D\beta} \tag{9}$$

**Table 2  GWO-parameters.**

| Parameter name | Range |
|---|---|
| Population size | 5 to 20 |
| alpha ($\alpha$) | 0 to 2 |
| beta ($\beta$) | 0 to 2 |
| delta ($\delta$) | 0 to 2 |
| omega ($\omega$ | 0 to 2 |
| Maximum iterations | 250 to 500 |

$$\vec{G3} = \vec{G\delta} - \vec{A3}.\vec{D\delta} \tag{10}$$

$$\vec{G}(t+1) = \frac{\vec{G1} + \vec{G2} + \vec{G3}}{3}. \tag{11}$$

During each iteration, omega wolves adjust their positions based on the positions of alpha, beta, and delta wolves, as these wolves possess superior knowledge regarding the potential location of prey.

## RESULTS ANALYSIS

For GWO-based parameter selection for GARCH and ARIMA models in stock price prediction for experimental work, consider Indian stock market data. The datasets include stock from indices such as Nifty and Bank Nifty and consider individual stock price data like Axis Bank, HFDC Bank, *etc.* The dataset encompasses a significant period, including 2008 and 2023.

The grey wolf optimizer algorithm is implemented using the R code. This algorithm aims to find the best order of parameter selection in ARIMA and GARCH models. Therefore, the GWO method is integrated into the parameter selection process for the GARCH and ARIMA models. To minimize the error in the forecasting model, we have incorporated the GWO method to find the best possible values for the parameters. The GWO method requires proper tuning, considering population size, the maximum number of iterations, and the search range. For experimental work, we have fine-tuned the GWO parameters, namely population size, alpha ($\alpha$), beta ($\beta$), delta ($\delta$), and omega ($\omega$). The detailed range values of GWO parameters are described in Table 2. For the proposed work experiment, we considered a population size of 20 and a maximum iteration of 500. Best search space parameters using the based method for HDFC Bank, SBIN Bank, Adani stock, and Infosys stock are described in Figs. 4, 5, 6 and 7. HDFC Bank's search space value is 4.75 for 50 iterations, and after that, its convergence. SBIN Bank search space value is 26.29 for 50 iterations; after that, it convergence. Adani stock search space value is 12.16 for 50 iterations, and after that, its convergence. Infosys stock search space value is 13.92 for 50 iterations, and after that, its convergence. These are the best search space parameters for the above stock using the GWO method. To validate the stock price prediction for the

```
$best
 [1] 4.751977 4.751977 4.751977 4.751977 4.751977 4.751977 4.751977 4.751977
 [9] 4.751977 4.751977 4.751977 4.751977 4.751977 4.751977 4.751977 4.751977
[17] 4.751977 4.751977 4.751977 4.751977 4.751977 4.751977 4.751977 4.751977
[25] 4.751977 4.751977 4.751977 4.751977 4.751977 4.751977 4.751977 4.751977
[33] 4.751977 4.751977 4.751977 4.751977 4.751977 4.751977 4.751977 4.751977
[41] 4.751977 4.751977 4.751977 4.751977 4.751977 4.751977 4.751977 4.751977
[49] 4.751977 4.751977
```

**Figure 4**  Best search space parameter for HDFC bank stock using GWO method.

```
$best
 [1] 26.2918 26.2918 26.2918 26.2918 26.2918 26.2918 26.2918 26.2918 26.2918
[10] 26.2918 26.2918 26.2918 26.2918 26.2918 26.2918 26.2918 26.2918 26.2918
[19] 26.2918 26.2918 26.2918 26.2918 26.2918 26.2918 26.2918 26.2918 26.2918
[28] 26.2918 26.2918 26.2918 26.2918 26.2918 26.2918 26.2918 26.2918 26.2918
[37] 26.2918 26.2918 26.2918 26.2918 26.2918 26.2918 26.2918 26.2918 26.2918
[46] 26.2918 26.2918 26.2918 26.2918 26.2918
```

**Figure 5**  Best search space parameter for SBIN bank stock using GWO method.

```
$best
 [1] 12.16518 12.16518 12.16518 12.16518 12.16518 12.16518 12.16518 12.16518
 [9] 12.16518 12.16518 12.16518 12.16518 12.16518 12.16518 12.16518 12.16518
[17] 12.16518 12.16518 12.16518 12.16518 12.16518 12.16518 12.16518 12.16518
[25] 12.16518 12.16518 12.16518 12.16518 12.16518 12.16518 12.16518 12.16518
[33] 12.16518 12.16518 12.16518 12.16518 12.16518 12.16518 12.16518 12.16518
[41] 12.16518 12.16518 12.16518 12.16518 12.16518 12.16518 12.16518 12.16518
[49] 12.16518 12.16518
```

**Figure 6**  Best search space parameter for Adani stock using GWO method.

GWO-GARCH model, we used a QQ (quantile–quantile) plot. The QQ plot evaluates the residuals of a GWO-GARCH model under the assumption of normality, and it is described in Figs. 8 and 9. The residuals are the values that are different from the GARCH model predictions. Quantiles of these residuals are plotted on a QQ plot and compared to theoretical quantiles, usually those of the standard normal distribution.

Novel GWO-based GARCH and ARIMA model performance is evaluated using RMSE, MSE, and MAPE metrics and defined in the Eqs. (12), (13) and (14). The GWO-GARCH model performs better than the GWO-ARIMA, ARIMA, and GARCH models. HDFC Bank GWO-GARCH model RMSE score is 0.13%, MSE score 0.17%, and MAE score is 0.14%. Axis Bank GWO-GARCH model RMSE score is 0.31%, MSE score 0.30%, and MAE score is 0.33%. SBIN Bank GWO-GARCH model RMSE score is 0.38%, MSE score 0.39% and MAE score is 0.44%. Adani stock GWO-GARCH model RMSE score is 0.45%, MSE score 0.49%, and MAE score is 0.48%. Infosys GWO-GARCH model RMSE score is 0.23%, MSE score 0.24% and MAE score is 0.27%. TCS GWO-GARCH model RMSE score is 0.25%, MSE score 0.24%, and MAE score is 0.27%. In experimental results for

```
$best
 [1] 13.92934 13.92934 13.92934 13.92934 13.92934 13.92934 13.92934 13.92934
 [9] 13.92934 13.92934 13.92934 13.92934 13.92934 13.92934 13.92934 13.92934
[17] 13.92934 13.92934 13.92934 13.92934 13.92934 13.92934 13.92934 13.92934
[25] 13.92934 13.92934 13.92934 13.92934 13.92934 13.92934 13.92934 13.92934
[33] 13.92934 13.92934 13.92934 13.92934 13.92934 13.92934 13.92934 13.92934
[41] 13.92934 13.92934 13.92934 13.92934 13.92934 13.92934 13.92934 13.92934
[49] 13.92934 13.92934
```

**Figure 7   Best search space parameter for Infosys stock using GWO method.**

stock price prediction, the GWO-based parameter selection technique for GARCH and ARIMA models exhibits enhanced performance, resulting in a 0.5% to 0.8% improvement compared to conventional GARCH and ARIMA models. It is described in Table 3, and the proposed work results are compared to the existing state of the art, and the model's performance is outperformed. The proposed model is robust to handle the nonlinear patterns in the data.

The variation in the results is due to volatility in the individual stock prices because stock prices have fluctuated with many factors like demand and supply of liquidity, earnings of the company, geopolitical tension, etc.,

$$RMSE = \sqrt{(\frac{1}{n})\sum_{i=1}^{n}(y_i-x_i)^2} \tag{12}$$

$$MSE = \frac{1}{n}\sum_{i=1}^{n}(y_i-x_i)^2 \tag{13}$$

$$MAE = (\frac{1}{n})\sum_{i=1}^{n}|y_i-x_i|. \tag{14}$$

## CONCLUSIONS

This work proposed a novel approach for predicting stock prices using the grey wolf optimizer (GWO) method to select parameters in GARCH and ARIMA models. The parameter selection in GARCH and ARIMA models is challenging due to stock price volatility. The study aimed to enhance prediction accuracy by effectively determining the optimal parameters for these models. The results reported in the research demonstrated the efficacy of the GWO algorithm in choosing appropriate GARCH and ARIMA model parameters. The prediction accuracy was significantly increased using the proposed method compared to the conventional methods that use arbitrary or human parameter selection. Combining the GWO-GARCH models improves the prediction model's accuracy

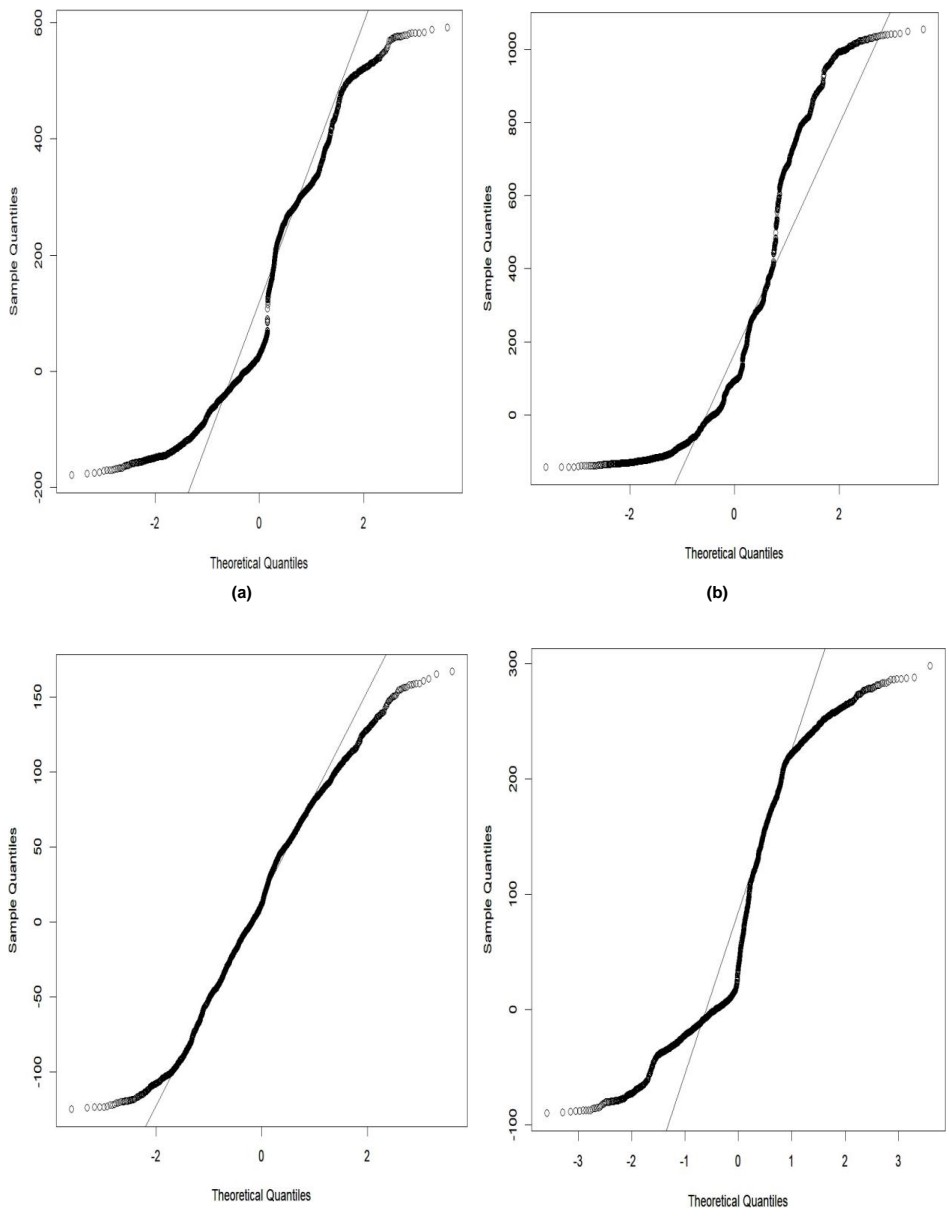

**Figure 8   Goodness of fit of GWO-GARCH model QQ plot.**

compared to GWO-ARIMA, GARCH, and ARIMA models. The limitation of GWO-based stock price prediction is that convergence speed may be better for stock prediction. The algorithm could become stuck in local optima or take longer to converge to effective solutions. Further research is necessary to finetune the GWO algorithm and explore its application in other forecasting models. With continuous advancements in optimization algorithms and machine learning techniques, the future of stock price prediction holds excellent potential for more accurate and reliable forecasting methods.

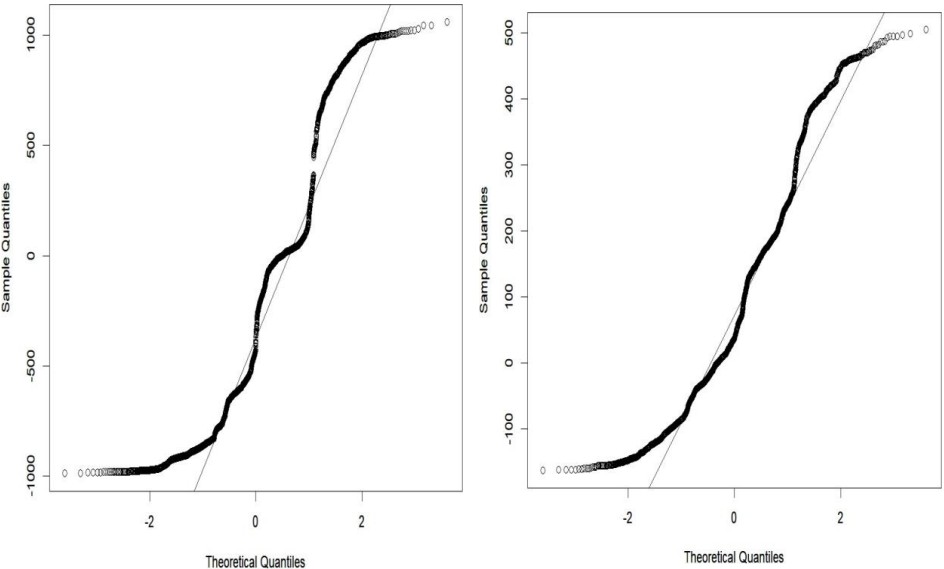

**Figure 9** **Goodness of fit of GWO-GARCH model QQ plot.**

## Abbreviations

| | |
|---|---|
| **ARIMA** | Autoregressive integrated moving average |
| **GARCH** | Generalized autoregressive conditional heteroskedasticity |
| **RMSE** | Root mean square error |
| **MSE** | Mean squared error |
| **GWO** | Grey wolf optimizer |
| **QQ** | Quantile-Quantile |
| **TCS** | Tata Consultancy Services |
| **SVM** | Support vector machine |
| **MAE** | Mean absolute error |
| **SBIN** | State Bank of India |
| **GIC** | Generalized information criteria |
| **DCC** | Dynamic conditional covariance |
| **SVR** | Support vector regression |
| **ENN** | Elman neural network |
| **DE** | Differential evolution network |
| **SVM** | Support vector machine |

## ACKNOWLEDGEMENTS

The authors acknowledge Editor-in-chief, Prof. Zeljko Stevic of this journal for the constant encouragement to finalize the article. Also, the authors are grateful for the comments and suggestions by the referee. Their comments and suggestions greatly improved the article.

**Table 3  Results comparison.**

| Stock name | Model | RMSE | MSE | MAE |
|---|---|---|---|---|
| HDFC Bank | ARIMA | 0.39 | 0.38 | 0.37 |
| HDFC Bank | GARCH | 0.28 | 0.31 | 0.34 |
| HDFC Bank | GWO-ARIMA | 0.21 | 0.23 | 0.25 |
| HDFC Bank | GWO-GARCH | 0.13 | 0.17 | 0.14 |
| Axis Bank | ARIMA | 0.44 | 0.48 | 0.46 |
| Axis Bank | GARCH | 0.41 | 0.43 | 0.45 |
| Axis Bank | GWO-ARIMA | 0.38 | 0.39 | 0.36 |
| Axis Bank | GWO-GARCH | 0.31 | 0.30 | 0.33 |
| SBIN Bank | ARIMA | 0.55 | 0.57 | 0.61 |
| SBIN Bank | GARCH | 0.50 | 0.52 | 0.54 |
| SBIN Bank | GWO-ARIMA | 0.45 | 0.47 | 0.49 |
| SBIN Bank | GWO-GARCH | 0.38 | 0.39 | 0.44 |
| Adani Stock | ARIMA | 0.63 | 0.67 | 0.65 |
| Adani Stock | GARCH | 0.59 | 0.60 | 0.62 |
| Adani Stock | GWO-ARIMA | 0.52 | 0.55 | 0.54 |
| Adani Stock | GWO-GARCH | 0.45 | 0.49 | 0.48 |
| Infosys | ARIMA | 0.36 | 0.39 | 0.38 |
| Infosys | GARCH | 0.30 | 0.33 | 0.35 |
| Infosys | GWO-ARIMA | 0.27 | 0.30 | 0.28 |
| Infosys | GWO-GARCH | 0.23 | 0.24 | 0.27 |
| TCS | ARIMA | 0.39 | 0.44 | 0.47 |
| TCS | GARCH | 0.35 | 0.40 | 0.42 |
| TCS | GWO-ARIMA | 0.31 | 0.34 | 0.34 |
| TCS | GWO-GARCH | 0.25 | 0.24 | 0.27 |
| SunPharma | ARIMA | 0.30 | 0.35 | 0.34 |
| SunPharma | GARCH | 0.29 | 0.32 | 0.33 |
| SunPharma | GWO-ARIMA | 0.27 | 0.29 | 0.33 |
| Sunpharma | GWO-GARCH | 0.23 | 0.25 | 0.26 |
| NatcoPharma | ARIMA | 0.49 | 0.52 | 0.54 |
| NatcoPharma | GARCH | 0.45 | 0.47 | 0.49 |
| NatcoPharma | GWO-ARIMA | 0.43 | 0.47 | 0.44 |
| NatcoPharma | GWO-GARCH | 0.40 | 0.39 | 0.44 |
| HDFC Bank | INC(U) *Singh et al. (2022)* | 0.60 | – | 0.13 |
| Infosys | INC(U) *Singh et al. (2022)* | 0.67 | – | 0.33 |
| TCS | INC(U) *Singh et al. (2022)* | 0.112 | – | 0.28 |

### Funding

The authors received no funding for this work.

### Competing Interests

The authors declare no conflict of interest.

## Author Contributions

- Sneha S Bagalkot conceived and designed the experiments, performed the experiments, analyzed the data, performed the computation work, prepared figures and/or tables, authored or reviewed drafts of the article, and approved the final draft.
- Dinesha H A conceived and designed the experiments, performed the computation work, authored or reviewed drafts of the article, and approved the final draft.
- Nagaraj Naik conceived and designed the experiments, performed the experiments, analyzed the data, performed the computation work, prepared figures and/or tables, authored or reviewed drafts of the article, and approved the final draft.

## Data Availability

The code is available in the Supplemental File.

The data is available at NSE Historical Reports: https://www.nseindia.com/resources/historical-reports-capital-market-daily-monthly-archives.

## Supplemental Information

Supplemental information for this article can be found online at http://dx.doi.org/10.7717/peerj-cs.1735#supplemental-information.

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
