# Peer review of "Novel grey wolf optimizer based parameters selection for GARCH and ARIMA models for stock price prediction"

_PeerJ Computer Science, doi:10.7717/peerj-cs.1735_

## Round 0.1 · original submission · Major Revisions

Dear authors,

Your paper has been reviewed by three reviewers with decisions: reject, major and minor. We want to give you chance to revise your paper, so please mark all changes and provide cover letter with replies point to point to each comment.

Reviewer 1 ·

Basic reporting

There is a growing classical concern about analyzing financial markets (stock market prices) in terms of time series and volatility and creating sophisticated models that can predict relatively well. Over the years, 'artificial neural networks', 'genetic algorithms', 'fuzzy-based techniques', and recently 'machine learning' has been widely used in this field. On the other hand, due to the dynamic and volatile nature of the stock market, more robust models that can accurately predict market prices need to be developed.
• Computational intelligence consists of various mechanisms to solve complex problems in different environments. This field may also involve using many alternative methods such as artificial neural networks, evolutionary computation, and swarm intelligence. Swarm intelligence is a type of computational intelligence used to solve optimization problems. These algorithms include many different types of algorithms. “Gray wolf optimization” is one of these alternative types.
• In this study, stock price estimation was made. Time series data were analyzed with the ARIMA model. The GARCH model, on the other hand, explains the time-varying volatility of financial data. However, in GARCH and ARIMA models, parameter selection is done by trial and error, which is a challenging and time-consuming task. Therefore, the GWO method has been taken into account for parameter selection in the proposed study.

Experimental design

• The fact that the adjustment of GWO parameters was not well-researched in the literature in this study was considered an important gap. In fact, parameter tuning is important for all optimization algorithms when solving real-world problems. Investigating and suggesting alternative equations for parameter adjustment would be a valuable contribution.

Validity of the findings

• On the other hand, wouldn't it be reasonable to compare some of the SI formats from a comparative perspective? Could this work demonstrate Gray Wolf Optimizer's ability in terms of discovery power compared to other metaheuristic algorithms? I am waiting for appropriate answers to this question.
• Scope of this research is limited to a small number of firms in India without covering the entire financial markets. The study describes the best search field parameters using the GWO-based method for HDFC Bank, SBIN Bank, Adani stock and Infosys stock. Making a generalization on only four stocks, in my opinion, reflects an imperfect point of view for forecasting. I suggest that the authors explain well the reasons for this or include more and a reasonable number of stocks in the analysis.

Reviewer 2 ·

Basic reporting

1. The abstract is too extended and does not coincide. Focus more on the problem statement or
objectives, (ii) research method and design, (iii) summary of major findings, and (iv) brief conclusions

2. The problem statement mentioned that "However, parameter selection in ARIMA and GARCH models is a challenging task because most of the work considered a manual method to select the parameters in ARIMA and GARCH models [31],[7]." Still, most software already has an automation package to estimate ARIMA.

3. Some parts of the English and the notations are not up to standard and are sometimes somewhat ambiguous.

4. The literature review is irrelevant to the research issue-automation parameter selection.

5. Figures and Tables should be put near the discussion.

Experimental design

1. The background, problem statement, and literature review do not identify the knowledge gap.

2. The proposed method should not be discussed based on analogy (grey wolves), but it should be based on data analysis.

3. The data analysis should start with testing for stationary of the data.

Validity of the findings

1. The study should also compare with the hybrid of ARIMA-GARCH.

2. Results of error measurements should be presented in decimals, not in percentages because the superiority of a method is measured based on smaller values.

3. The findings should be discussed with appropriate references.

Additional comments

No comment.

·

Basic reporting

I think this work need more work . But I appreciate the application part.
Comments:
Extended the abstract and conclusion which add perspectives section.
Add some recent references in introduction and motivation section.
Correct the misprint and grammar mistakes.
Add some graphics in application part.
Add discussion section.

Experimental design

Add graphics for more explication.
Add more models for comparison section.

Validity of the findings

In conclusion add perspectives of biliniear models.

Additional comments

Add references like:
. On Modelling seasonal ARIMA series: Comparison, Application and Forecast (Number of Injured in Road Accidents in Northeast Algeria).
. On volatility swaps for stock market forecast: Application example CAC 40 French index.
. Modelling of oil price volatility using ARIMA-GARCH models.
. Benefit of GARCH multivariate models: application to the energy market.

---

## Round 0.2 · Minor Revisions

Dear author,

Please revise the paper again according to comments by reviewer 3.

Reviewer 1 ·

Basic reporting

In your opinion, "The effectiveness of automated algorithms in capturing data complexities is sometimes limited, negatively affecting the optimal selection of model parameters". At this point, you can write your insightful recommendations regarding the development of automated algorithms for stock selection. Therefore, it would be useful to highlight the findings of the study. I recommend adding these comparative suggestions to the abstract, introduction, literature, and discussion sections.

Experimental design

The experimental design is well structured.

Validity of the findings

no comment

Additional comments

no comment

Reviewer 2 ·

Basic reporting

The authors failed to adequately address or consider some comments from earlier reviews.

Experimental design

The authors failed to adequately address or consider some comments from earlier reviews.

Validity of the findings

The authors failed to adequately address or consider some comments from earlier reviews.

Additional comments

The authors failed to adequately address or consider some comments from earlier reviews.

·

Basic reporting

Dear professor,
I think now this work is suitable to published in PeerJ Computer Science. But I have some minor comments:
- correct the misprint and gramatical errors.
- figues 8 and 9 are not clear, improve it.
- add Acknowledgements such as:Dr. The authors acknowledge Editor-in-chief, Prof. ..........., of this journal for the constant encouragement to finalize the paper. Also, the authors are grateful for the comments and suggestions by the referee. Their comments and suggestions greatly improved the article.

Experimental design

good

Validity of the findings

good work

Additional comments

Format of the paper must change according to journal template.

---

## Round 0.3 · accepted · Accept

Dear authors,

Your revised version of the paper has been accepted by two of three reviewers. The third reviewer did not evaluate the revisions at this stage, so the evaluation has been made by the Academic Editor who believes that the paper has enough quality to be accepted.

·

Basic reporting

After revision, I can accept this paper .

Experimental design

Good

Validity of the findings

Statistically sound.

Additional comments

Check the grammar and orthograph errors.